# A Multimodal BiMamba Network with Test-Time Adaptation for Emotion Recognition Based on Physiological Signals

**Ziyu Jia**[1,2]    **Tingyu Du**[3,4]    **Zhengyu Tian**[5]    **Hongkai Li**[5]    **Yong Zhang**[6*]    **Chenyu Liu**[7*]
[1]Institute of Automation, Chinese Academy of Sciences
[2]Shanghai Key Laboratory of Data Science
[3]Beijing Key Laboratory of Mobile Computing and Pervasive Devices,
Institute of Computing Technology, Chinese Academy of Sciences
[4]University of Chinese Academy of Sciences
[5]School of Computer Science and Technology, Beijing Jiaotong University
[6]School of Information Engineering, Huzhou University
[7]College of Computing and Data Science, Nanyang Technological University
jia.ziyu@outlook.com,
{tingyu_du, zhengyu_tian, hongkaili7453}@sina.com,
zhyong@zjhu.edu.cn, chenyu003@e.ntu.edu.sg

## Abstract

Emotion recognition based on physiological signals plays a vital role in psychological health and human–computer interaction, particularly with the substantial advances in multimodal emotion recognition techniques. However, two key challenges remain unresolved: 1) how to effectively model the intra-modal long-range dependencies and inter-modal correlations in multimodal physiological emotion signals, and 2) how to address the performance limitations resulting from missing multimodal data. In this paper, we propose a multimodal bidirectional Mamba (Bi-Mamba) network with test-time adaptation (TTA) for emotion recognition named BiM-TTA. Specifically, BiM-TTA consists of a multimodal BiMamba network and a multimodal TTA. The former includes intra-modal and inter-modal BiMamba modules, which model long-range dependencies along the time dimension and capture cross-modal correlations along the channel dimension, respectively. The latter (TTA) mitigates the amplified distribution shifts caused by missing multimodal data through two-level entropy-based sample filtering and mutual information sharing across modalities. By addressing these challenges, BiM-TTA achieves state-of-the-art results on two multimodal emotion datasets.

## 1 Introduction

The mining and analysis of emotional data contribute to the diagnosis of mental disorders and psychological health assessment. In recent years, data sources for emotion recognition research have mainly focused on two aspects: non-physiological signals and physiological signals[1]. Non-physiological signals, such as audio and video, are affected by subjective bias and intentional concealment[2]. This makes it difficult for non-physiological signals to represent the emotional state of the body reliably. In contrast, physiological signals, such as electroencephalography (EEG) and electrooculography (EOG), reflect the true emotional state of the human body objectively[3–

---

[*]Corresponding author.

39th Conference on Neural Information Processing Systems (NeurIPS 2025).

6]. Therefore, emotion recognition based on physiological signals demonstrates great application potential in fields related to mental health[7, 8].

Moreover, compared to unimodal physiological signals, emotion recognition based on multimodal physiological signals combines information from multiple sources to provide a more comprehensive emotional assessment. Although multimodal physiological methods for emotion recognition have achieved significant progress[9–11], there are still two challenges:

1) *How to effectively model both the intra-modal long-range dependencies and inter-modal correlations of multimodal emotion-related physiological signals.* For the former, as a typical form of time-series data, emotion-related physiological signals exhibit long-range dependencies reflecting the gradual accumulation of emotional changes. Emotions, such as anxiety, develop progressively and require a duration for their physiological manifestations to accumulate. Such accumulation underscores that emotional changes are not just a state that occurs instantaneously but a process that evolves. For the latter, the inter-modal correlations are evident in the way different modalities respond. For instance, increases in EEG activity during emotional arousal are often accompanied by galvanic skin response (GSR) conductance peaks and decreases in electrocardiogram (ECG) heart-rate variability[12].

Traditional backbone networks have limitations in modeling intra-modal long-range dependencies and inter-modal correlations. For intra-modal modeling, CNN-based networks[13, 14] excel at extracting local features but struggle to capture essential long-range temporal information. Although stacking more convolutional layers together with pooling operations to expand the receptive field can theoretically increase context[15–17], it often degrades fine-grained local details[18]. What's more, Transformer-based networks are able to capture global emotional patterns, but their attention mechanism lacks explicit temporal filtering capability and tends to distribute weight uniformly across all time steps. This makes it difficult for these methods to construct practical long-range dependencies based on key nodes of emotion fluctuations. For inter-modal modeling, the token-level attention in Transformer-based networks only computes pairwise relationships[19] between channels and maintains only instantaneous interaction states. However, the complex dependencies inherent in physiological signals cannot be fully captured through pairwise channel-wise interactions, making it difficult to model high-order correlations. This limitation essentially stems from the lack of explicit global state variables in the attention mechanism[20], which hinders the effective integration of cross-modal information. Therefore, how to effectively model the intra-modal long-range dependencies and inter-modal correlations of multimodal physiological signals remains a significant challenge.

2) *How to mitigate the impact of missing multimodal emotion-related physiological data on model performance.* When acquiring emotion-related physiological signals, subjects must remain seated for extended periods while exposed to high-intensity stimuli to generate emotional responses. However, uncontrollable factors such as perspiration, changes in posture, or the drying of the conductive gel may cause sensors to slip or experience poor contact, resulting in incomplete multimodal signal acquisition to varying degrees[21, 22]. Such incompleteness amplifies the distribution shifts in emotion-related physiological data, ultimately leading to degraded model performance. First, emotion-related physiological data inherently exhibit distribution shifts between the training and test sets[23]. These shifts arise from within-subject variations in emotional state, cognitive load, and environmental conditions across sessions. Furthermore, the unavoidable presence of missing data disrupts the original data patterns, skews feature distributions, and introduces bias, which collectively amplify the existing distribution shifts in physiological data. This amplification makes it increasingly challenging for pre-trained models to capture emotional patterns accurately.

Existing methods focus on training phase measures to deal with the missing data problem. For example, Salazar et al. [24] propose feature- and decision-level fusion methods to address the data missingness issue in multimodal emotion recognition. However, these methods typically require retraining to adapt to new missing data patterns, limiting their application flexibility. Furthermore, a promising approach is to mitigate the effects of missing data by fine-tuning the model during the testing phase. For instance, Yang et al. [25], Lei and Pernkopf [26] address unimodal corruption in multimodal data by adjusting self-attention modules to assign lower weights to the corrupted modality. However, these methods cannot handle cases of multimodal corruption where multiple physiological signals are simultaneously missing. Therefore, mitigating the negative impact of missing multimodal data on emotion recognition models remains a significant challenge in practical applications.

In light of these challenges, Mamba's selective input mechanism efficiently models long-range dependencies, and its explicit global state variables as part of its state space model (SSM) capture inter-modal correlations simultaneously, thereby effectively addressing the first challenge. Additionally, test-time adaptation (TTA) requires only minimal parameter fine-tuning to mitigate distribution shifts, thus resolving the second challenge. Therefore, we propose a multimodal bidirectional Mamba (BiMamba) network with TTA for emotion recognition named BiM-TTA. BiM-TTA consists of a multimodal BiMamba network and a multimodal TTA. The multimodal BiMamba network includes an intra-modal BiMamba module and an inter-modal BiMamba module, and the multimodal TTA includes two-level entropy-based sample filtering and mutual information sharing across modalities.

Overall, the key contributions of this study can be summarised as follows:

- We design a multimodal BiMamba network, where the intra-modal BiMamba module models long-range dependencies within modalities, and the inter-modal BiMamba module captures inter-modal correlations.

- We propose a multimodal TTA method that alleviates the negative impact of amplified distribution shifts caused by missing multimodal data on model performance.

- The evaluation of BiM-TTA on two multimodal emotion datasets confirms its state-of-the-art performance and effectiveness.

## 2    Related Work

In recent years, physiological signals have become a key focus in emotion recognition due to their ability to accurately and objectively reflect the genuine emotions of subjects. For instance, Liu et al. [27] used maximum relevance and minimum redundancy to extract emotional information for feature selection. Similarly, Bazgir et al. [28] enhanced the accuracy by applying a cross-validated SVM with a radial basis function kernel for classification. However, traditional machine learning methods, which often rely heavily on expert knowledge, are limited by feature design and selection [29, 30].

To overcome these limitations, researchers have applied deep learning methods to emotion recognition tasks. For example, in unimodal emotion recognition, Jia et al. [31] proposed SST-EmotionNet, an attention-based 3D dense network that simultaneously integrates spatial-spectral-temporal features within a unified framework. Ding et al. [14] designed TSception, a multi-scale convolutional neural network that extract temporal dynamics and spatial asymmetry features from EEG signals. These deep learning methods achieve remarkable results in the unimodal domain.

Moreover, studies have shown that multimodal methods can better capture the diversity and complexity of emotions, resulting in superior performance in emotion recognition. For instance, Ma et al. [32] developed a multimodal residual LSTM network, sharing weights across modalities. Hssayeni and Ghoraani [33] explored the use of deep convolutional neural networks for two multimodal data fusion to evaluate positive and negative emotions.Koorathota et al. [34] proposed the Multimodal Neurophysiological Transformer, which employs cross-modal attention to capture inter-modal correlations. Chen et al. [35] designed an attention-based recurrent graph convolutional network that integrates multimodal physiological features with a convolutional block attention module. Huang et al. [36] introduced GJFusion to address modality heterogeneity through channel-level inter-modality correlations based on graph joints.

Despite achieving promising performance, existing methods in multimodal emotion recognition primarily rely on traditional backbone networks, posing limitations in modeling long-range dependencies and inter-modal correlations of multimodal physiological signals related to emotion. In addition, these methods rarely address the issue of missing multimodal data. To address these challenges, we propose BiM-TTA, which consists of the multimodal BiMamba backbone network and TTA. Further discussion of related works on Mamba and TTA is presented in Appendix A.1.

## 3    Methodology

As shown in Figure 1, we propose BiM-TTA, comprising the multimodal BiMamba network and the multimodal TTA. The multimodal BiMamba network consists of intra- and inter-modal BiMamba modules, which extract modality-specific features and perform feature fusion across modalities.

The multimodal TTA includes two steps: two-level entropy-based sample filtering and mutual information sharing across modalities. These steps effectively mitigate the impact of distribution shifts amplification caused by missing multimodal data on model performance.

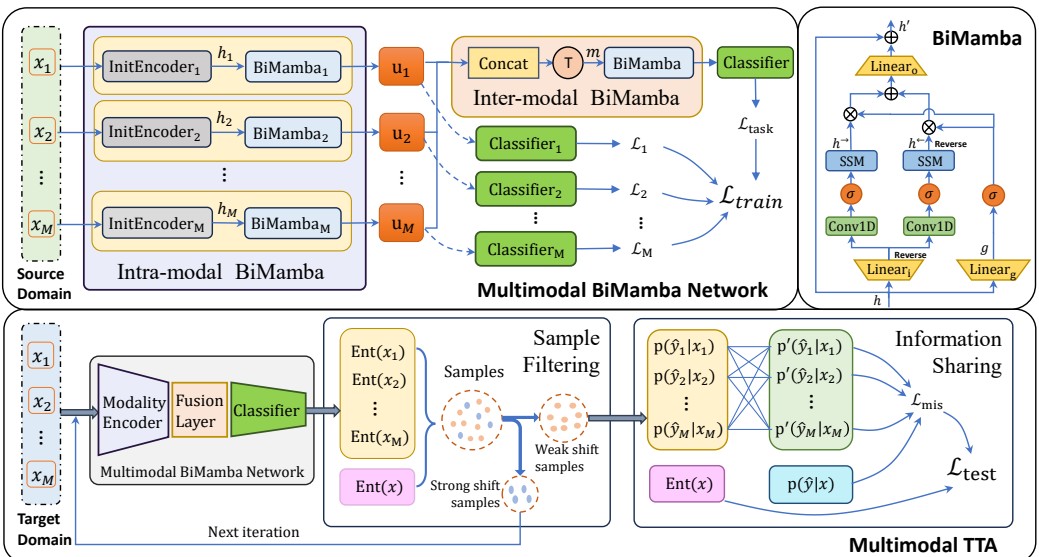

Figure 1: The overall structure of BiM-TTA. It contains the multimodal BiMamba network and the multimodal TTA. For the former, the intra- and inter-modal BiMamba modules capture and fuse the features of the different modalities, respectively. For the latter, samples with weak distribution shifts and rich multimodal information are selected for adaptation, while the remaining samples are retained until the next iteration. Then, mutual information sharing across modalities is performed to match the information between different modalities effectively. "Concat" represents the concatenation of multiple feature vectors $u_1, u_2, \ldots, u_M$. "T" denotes matrix transposition.

## 3.1 Preliminary

In this paper, we define emotion recognition based on physiological signals as a multi-class classification task for time-series data. The input data consists of physiological signals $x = \{x_1, x_2, \ldots, x_M\}$, where $x_i \in \mathbb{R}^{C_i \times L}, i \in \{1, 2, \ldots, M\}$ represents the $i$-th-modality data, $C_i$ is the number of acquisition channels for the $i$-th modality, and $L$ represents the number of time steps of the input signal. The ground truth label $y$ is a discrete value corresponding to a category, which is then converted into a one-hot encoding $p(y \mid x) \in \mathbb{R}^N$, where $N$ is the number of categories. The model outputs the predicted category probability $p(\hat{y} \mid x) \in \mathbb{R}^N$.

## 3.2 Multimodal BiMamba Network

As shown in the upper part of Figure 1, the multimodal BiMamba network consists of two modules: the intra-modal BiMamba module and the inter-modal BiMamba module. The former is designed to model each modality independently, and the latter effectively fuses multimodal features.

### 3.2.1 Intra-modal BiMamba Module

Mamba is designed for natural language processing tasks, where its output at each time step depends only on the current input and hidden state, without involving information from future time steps. This unidirectional modeling approach is well suited for generative autoregressive tasks, as these tasks rely primarily on previous information for inference[37]. However, in emotion classification tasks based on physiological signals, it is necessary to capture contextual information from physiological time series simultaneously, and a unidirectional approach cannot fully capture the complex temporal patterns. To address this limitation, we design the intra-modal BiMamba module. This module comprehensively integrates the temporal features of each modality through a selective input mechanism and bidirectional modeling.

Initially, we design the initial encoder $\text{InitEncoder}_i(\cdot)$ to extract shallow features from a specific modality:

$$h_i = \text{InitEncoder}_i(x_i), i \in \{1, 2, \ldots, M\} \tag{1}$$

where $x_i \in \mathbb{R}^{C_i \times L}$ is the input of the $i$-th modality, in which $C_i$ denotes the number of input channels for the $i$-th modality and $L$ denotes the number of time steps. $h_i \in \mathbb{R}^{L' \times C_i'}$ represents the initial feature representation output of the $i$-th modality, $C_i'$ represents the number of output channels of the $i$-th modality, and $L'$ represents the feature length of the output signal. $M$ represents the total number of modalities.

Next, we introduce the BiMamba to model the temporal dimension further, and its structure is shown in Figure 1. Its core consists of the following three steps:

1) First, BiMamba employs a gating mechanism that adaptively weights features to highlight emotion-relevant information while suppressing noise and redundancy:

$$g_i = \sigma\left(W_i^g h_i + b_i^g\right) \tag{2}$$

where $g_i$ represents the output of the gating mechanism. $\sigma(\cdot)$ represents the $SiLU$ activation function. $W_i^g$ represents the weight matrix and $b_i^g$ represents the bias.

2) Second, BiMamba models temporal dependencies from both forward and backward perspectives using state-space modeling, thereby capturing richer contextual dynamics:

$$h_i^{\rightarrow} = g_i \otimes \text{SSM}_{\rightarrow}\left(\sigma\left(\text{Conv1D}_{\rightarrow}\left(W_i^h h_i + b_i^h\right)\right)\right) \tag{3}$$

$$h_i^{\leftarrow} = g_i \otimes \text{SSM}_{\leftarrow}\left(\sigma\left(\text{Conv1D}_{\leftarrow}\left(\text{Rev}_t\left(W_i^h h_i + b_i^h\right)\right)\right)\right) \tag{4}$$

where $h_i^{\rightarrow} \in \mathbb{R}^{L' \times C_i'}$ and $h_i^{\leftarrow} \in \mathbb{R}^{L' \times C_i'}$ are the feature representations of the $i$-th modality in forward and backward modeling. $\text{Rev}_t(\cdot)$ denotes the flipping on the time dimension $L'$. SSM represents Selective State Space Model. $W_i^h$ represents the weight matrix and $b_i^h$ represents the bias.

3) Finally, BiMamba applies linear projection and residual connection to integrate bidirectional features and stabilize training:

$$u_i = h_i + \left(W_i^o\left(\frac{h_i^{\rightarrow} + \text{Rev}_t\left(h_i^{\leftarrow}\right)}{2}\right) + b_i^o\right) \tag{5}$$

where $u_i \in \mathbb{R}^{L' \times C_i'}$ is the output of the intra-modal BiMamba module of the $i$-th modality. $W_i^o$ is the weight matrix, and $b_i^o$ is the bias.

### 3.2.2 Inter-modal BiMamba Module

In the intra-modal BiMamba module, we focus on state modeling at the unimodal time scale. However, relying solely on intra-modal modeling neglects inter-modal correlations that are crucial for comprehensive multimodal understanding. Hence, we propose the inter-modal BiMamba module for modeling correlations across modalities. Specifically, we concatenate the features of each modality along the channel dimension $C_i', i \in \{1, 2, \ldots, M\}$, and swap the time and channel dimensions. This process is defined as:

$$m = \text{Transpose}\left(u_1 \| u_2 \| \ldots \ldots \| u_M\right) \tag{6}$$

where $m \in \mathbb{R}^{\sum_{i=1}^{M} C_i' \times L'}$ represents the concatenated multimodal feature matrix. $\|$ represents concatenation along the channel dimension. $\text{Transpose}(\cdot)$ represents the swapping of the time and channel dimensions. This operation integrates the features of each modality into the unified feature matrix $m$. In $m$, each set of continuous channels represents the features of a modality.

Subsequently, we input the multimodal feature matrix $m$ into the BiMamba structure to perform bidirectional state modeling of the features from different modalities along the channel dimension $\sum_{i=1}^{M} C_i'$. This process is defined as:

$$H = \text{BiMamba}(m) \tag{7}$$

where $H \in \mathbb{R}^{\sum_{i=1}^{M} C_i' \times L'}$ represents the inter-modal feature fusion representation. $\text{BiMamba}(\cdot)$ is defined in 3.2.1. Through the inter-modal BiMamba module, the features of different modalities can interact with each other. In the forward process, the hidden states built from modalities

$\{1, 2, \ldots, i - 1\}$ assist in constructing the feature representation of the $i$-th modality. In the backward process, the hidden states derived from modalities $\{M, M - 1, \ldots, i + 1\}$ facilitate the construction of supplementary feature representations for the $i$-th modality. The bidirectional state modeling mechanism can capture inter-modal correlations from multiple perspectives, thereby enhancing the representational ability of multimodal features.

### 3.2.3 Auxiliary task

To effectively learn the optimal states of each modality encoder during training, while balancing the training progress across modalities and preventing overfitting in any single modality, we further design an auxiliary task. After independently modeling the unique feature representations of each modality through the intra-modal BiMamba module, an additional classifier is introduced to output unimodal prediction probabilities:

$$p\left(\hat{y}_i \mid x_i\right) = p\left(\hat{y}_i \mid u_i\right) = \text{softmax}\left(W_i u_i + b_i\right) \tag{8}$$

where $p(\hat{y}_i \mid x_i) \in \mathbb{R}^N$ represents the prediction probability of the $i$-th modality. $N$ is the number of classes. $u_i$ represents the feature matrix output by the intra-modal BiMamba module of the $i$-th modality. $W_i$ is the weight matrix and $b_i$ is the bias.

To optimize the auxiliary classifier for each modality, the cross-entropy loss $\mathcal{L}i$ is computed as:

$$\mathcal{L}_{\text{i}} = -\frac{1}{n} \sum_{j=1}^{n} p(y \mid x)^{(j)} \log p(\hat{y}_i \mid x_i)^{(j)} \tag{9}$$

where $p(y \mid x)^{(j)}$ represents the one-hot encoding of the label for the $j$-th sample classification task. $p(\hat{y}_i \mid x_i)^{(j)}$ represents the prediction probability of the $i$-th modality for the $j$-th sample. $n$ denotes the number of samples in a batch.

To jointly optimize the model across modalities, the overall training objective $\mathcal{L}_{train}$ is defined as the sum of the main classification loss $\mathcal{L}_{task}$ and the auxiliary losses from all modalities:

$$\mathcal{L}_{train} = \mathcal{L}_{task} + \sum_{i=1}^{M} \alpha_i \mathcal{L}_i \tag{10}$$

where $\mathcal{L}_{task}$ denotes the cross-entropy loss for the final classification output. $\alpha_i$ represents the auxiliary task weight of the $i$-th modality.

## 3.3 Multimodal TTA

As shown in the bottom half of Figure 1, the multimodal TTA method consists of two key steps: two-level entropy-based sample filtering and mutual information sharing across modalities.

### 3.3.1 Two-level Entropy-based Sample Filtering

Missing multimodal data affects samples differently, with a small subset suffering severe degradation of multimodal information due to the loss of key emotional cues. Directly using such samples for fine-tuning often harms model performance. To address this, we propose a two-level entropy-based sample filtering method:

(a) **Multimodal entropy** reflects the model's certainty about its prediction [38]. A lower value suggests that the prediction is more reliable, which often corresponds to a distribution in the target domain closer to that in the source domain. Using such samples for fine-tuning leads to a more stable and reliable model adaptation. Conversely, a higher value indicates greater uncertainty, which often corresponds to strong distribution shifts from the source domain, making these samples unsuitable for fine-tuning.

(b) **Unimodal entropy** measures the extent to which a sample relies on multimodal information [26, 39]. A lower value indicates that the prediction can be made primarily based on a single dominant modality, suggesting that the sample contains limited multimodal information. In contrast, a higher value implies that the prediction is more likely to integrate multiple modalities, indicating that the sample possesses richer multimodal information and is more suitable for fine-tuning.

Therefore, we design a method that selectively retains samples with low multimodal entropy and high unimodal entropy. These samples are more conducive to stable model adaptation and contain richer multimodal information. First, we compute the multimodal entropy $\text{Ent}(x)$ and unimodal entropy $\text{Ent}(x_i)$ of each sample:

$$\text{Ent}(x) = -\sum_{c=1}^{N} p(\hat{y} = c \mid x) \log p(\hat{y} = c \mid x) \tag{11}$$

$$\text{Ent}(x_i) = -\sum_{c=1}^{N} p(\hat{y}_i = c \mid x_i) \log p(\hat{y}_i = c \mid x_i) \tag{12}$$

where $p(\hat{y} \mid x)$ is the multimodal prediction probability, $p(\hat{y}_i \mid x_i)$ is the prediction probability of the $i$-th modality, and $N$ is the number of categories. Next, we employ an iterative entropy-based sample selection strategy to progressively expand the range of target-domain samples. During this process, the model first adapts to samples with weak distribution shifts and rich multimodal information. As the iterations progress, it gradually incorporates samples with strong distribution shifts to achieve smooth adaptation from the source domain to the target domain. Specifically, a gradually increasing threshold is used to ensure a smooth adaptation. The thresholds $\gamma_m$ and $\gamma_u$ defined as:

$$\gamma_m = \frac{1}{n} \sum_{j=1}^{n} \text{Ent}(x)^{(j)} + \gamma_m' * \beta_t \tag{13}$$

$$\gamma_u = \frac{1}{n} \sum_{j=1}^{n} \sum_{i=1}^{M} \mu_i \text{Ent}(x_i)^{(j)} - \gamma_u' * \beta_t \tag{14}$$

where $\gamma_m'$ and $\gamma_u'$ are related to the variances of multimodal and unimodal entropy, respectively. $\beta_t$ represents the smoothing factor. $\beta_t = \beta + \frac{t}{iter}(1 - \beta)$, $t$ is the current iteration number, $iter$ is the total number of iterations, $\beta$ is a hyperparameter. $M$ is the total number of modalities. $n$ represents the number of samples in a batch. Samples used in previous iterations are excluded from further ones. Details about the rationale for threshold selection are provided in Appendix A.2. Finally, the filtering employs the following identification criteria:

$$S(x) = \left\{ x \mid \text{Ent}(x) \leq \gamma_m \text{ and } \sum_{i=1}^{M} \mu_i \text{Ent}(x_i) \geq \gamma_u \right\} \tag{15}$$

where $\mu_i$ is the hyperparameter for the unimodal entropy weight of the $i$-th modality.

### 3.3.2 Mutual Information Sharing Across Modalities

Mutual information sharing across modalities uses the more informative modalities to guide the learning of modalities with significant missing information. This alleviates the impact of amplified distribution shifts between modalities, thereby mitigating the negative effects of amplified overall distribution shifts in the target domain. Specifically, we define the complementary probability of the prediction probability $p(\hat{y}_i \mid x_i)$ of the $i$-th modality as $p'(\hat{y}_i \mid x_i)$:

$$p'(\hat{y}_i \mid x_i) = \frac{\sum_{j=1}^{M} p(\hat{y}_j \mid x_j) - p(\hat{y}_i \mid x_i)}{M - 1} \tag{16}$$

where $M$ represents the total number of modalities.

To improve the consistency of predictions across different modalities, we can minimize the KL divergence between the probability $p(\hat{y}_j \mid x_j)$ and its complementary probability $p'(\hat{y}_i \mid x_i)$. However, if a modality is severely corrupted, minimizing the KL divergence may negatively impact the informative modalities. Therefore, we also include the multimodal probability $p(\hat{y} \mid x)$ to improve stability and reliability. The mutual information sharing loss across modalities $\mathcal{L}_{\text{mis}}(x)$ is defined as:

$$
\begin{aligned}
\mathcal{L}_{\text{mis}}(x) &= \sum_{i=1}^{M} D_{KL}\left( p(\hat{y}_i \mid x_i) \,\|\, \frac{1}{2}\left(p'(\hat{y}_i \mid x_i) + p(\hat{y} \mid x)\right) \right) \\
&= \sum_{i=1}^{M} \sum_{j=1}^{N} p(\hat{y}_i \mid x_i)^{(j)} \log \frac{2p(\hat{y}_i \mid x_i)^{(j)}}{p'(\hat{y}_i \mid x_i)^{(j)} + p(\hat{y} \mid x)^{(j)}}
\end{aligned}
\tag{17}
$$

where the factor $\frac{1}{2}$ represents the average weight. $N$ represents the number of prediction categories. Through mutual information sharing across modalities, the model can effectively align information between different modalities. When some modalities perform poorly, other modalities can provide valuable information, enhancing the overall prediction performance.

### 3.3.3 TTA optimization

Regarding the TTA optimization, we focus on two aspects: loss computation and fine-tuning details.

**Loss Computation**    We use weighted terms to emphasize the contribution of samples during the adaptation process. The weight term $\alpha(x)$ is defined as:

$$\alpha(x) = \frac{1}{\exp(\text{Ent}(x) - \text{Ent}_0)} \tag{18}$$

where $\text{Ent}_0$ represents the predefined normalization factor[40]. $\text{Ent}(x)$ is the multimodal entropy of the sample. Based on the weighted term, the final loss $\mathcal{L}_{test}$ is formulated as:

$$\mathcal{L}_{test}(x) = \alpha(x)\,\mathbb{I}_{\{x \in S(x)\}}(\text{Ent}(x) + \lambda\mathcal{L}_{mis}(x)) \tag{19}$$

where $\mathbb{I}_{\{\cdot\}}(\cdot)$ represents the indicator function. $S(x)$ is the retained sample set obtained through two-level entropy-based sample filtering. $\lambda$ represents the hyperparameter.

**Fine-Tuning Details**    Inspired by surgical fine-tuning[41], only the parameters of the first convolutional layer in the initial encoder, the first fully connected layer of the inter-modal BiMamba module, and all batch normalization layers are fine-tuned. This selective fine-tuning strategy effectively reduces computational overhead while maintaining the stability of the model's essential component. Finally, the objective function for multimodal TTA is defined as:

$$\min_{\hat{\Theta}} \mathcal{L}_{test}(x) \tag{20}$$

where $\hat{\Theta} \in \Theta$ represents the tunable parameters. $\Theta$ represents all the parameters of the model. This means that only the tunable parameters are updated during test based on $\mathcal{L}_{test}$, while the other parameters remain fixed. The details of the multimodal TTA algorithm are presented in Appendix A.3.

## 4 Experiments

### 4.1 Datasets

BiM-TTA is evaluated on two publicly available multimodal datasets: DEAP[42] and MAHNOB-HCI[43]. The detailed description of DEAP and MAHNOB-HCI is presented in Appendix A.4.

### 4.2 Experiment Settings and Implementation

We employ trial-wise 10-fold cross-validation experimental setups under two conditions: 1) without missing multimodal data, and 2) with missing multimodal data. We split each trial into 4s non-overlapping segments, also known as cropped experiments. In each round of the 10-fold cross-validation, 9 folds are used for training and the remaining fold is used for testing. Within the 9 training folds, 80% of the data are used for model training and the remaining 20% for validation. To ensure the representativeness of the evaluation, we make sure that the same trial does not appear in both the training and test sets, thereby avoiding potential data leakage risks [14]. In the experimental evaluation of missing multimodal data, we emulate varying degrees of data absence by applying random masking with predefined ratios. Specifically, masking ratios of 0.2, 0.4, 0.6, and 0.8 are used to represent increasing levels of data incompleteness, from mild to severe.

For modality selection, EEG, EOG, and electromyography (EMG) are used for DEAP, whereas EEG, ECG, and GSR are used for MAHNOB-HCI. The proposed BiM-TTA model is implemented on the PyTorch framework and optimized using Adam with a learning rate of 0.001. The batch size is configured to 64 for DEAP and 32 for MAHNOB-HCI. For a fair comparison, we modify all baseline methods to their multimodal versions. Detailed values of the hyperparameters mentioned in the paper are provided in Appendix A.5.

Our baseline includes two categories: the emotion recognition baseline models and the TTA baseline methods. Detailed descriptions of the baselines are in Appendix A.6.

## 4.3 Experiment Analysis

The proposed method is first evaluated against representative emotion recognition models on two datasets without missing data, and subsequently compared with state-of-the-art TTA methods with missing data. It achieves superior performance in both comparisons.

Table 1: Comparison of emotion recognition baselines on DEAP and MAHNOB-HCI in terms of valence and arousal accuracy.

| | DEAP | | MAHNOB-HCI | |
| --- | --- | --- | --- | --- |
| | Valence | Arousal | Valence | Arousal |
| SVM | 0.552 | 0.584 | 0.564 | 0.573 |
| EEGNet | 0.566 | 0.593 | 0.609 | 0.612 |
| ACRNN | 0.609 | 0.638 | 0.610 | 0.613 |
| HetEmotionNet | 0.625 | 0.633 | 0.607 | 0.601 |
| TSception | 0.613 | 0.635 | 0.633 | 0.599 |
| LGGNet | 0.618 | 0.636 | 0.632 | 0.609 |
| EEG-Deformer | 0.609 | 0.630 | 0.587 | 0.595 |
| MambaFormer | 0.621 | 0.587 | 0.588 | 0.619 |
| SST | 0.613 | 0.623 | 0.606 | 0.616 |
| VSGT | 0.631 | 0.628 | 0.613 | 0.599 |
| **BiM-TTA(ours)** | **0.673** | **0.641** | **0.650** | **0.635** |

For experiments without missing data, as shown in Table 1, deep learning methods generally outperform SVM. EEGNet and TSception primarily use CNNs to extract intra-modal temporal features. ACRNN combines CNNs and unidirectional LSTM to model temporal features, and HetEmotionNet uses unidirectional GRU to model the time dimension. However, these CNN-based and unidirectional RNN-based models have limitations in capturing long-range dependencies within the modality. Furthermore, EEG-Deformer combines CNNs and Transformers to extract complex temporal features. MambaFormer embeds Mamba modules into Transformer feed-forward layers to unify long- and short-range modeling, and SST couples a Mamba-based global expert with a windowed Transformer local expert and fuses their outputs. Although these hybrid models strengthen temporal dependency modeling, they still overlook inter-channel interactions, as they rely on simplified fusion strategies such as single-layer convolutions or linear projections, which constrain their capacity to capture complex inter-modal relationships. VSGT and LGGNet model the inter-channel relationships using GNNs, yet their temporal modeling relies solely on CNNs and fully connected layers, which hampers the capture of complex long-range dependencies. In contrast, BiM-TTA models long-range dependencies within modalities and correlations between modalities through the intra-modal BiMamba module and inter-modal BiMamba module. It achieves the best performance on both datasets, demonstrating the superiority of the multimodal BiMamba backbone network.

Table 2: Comparative analysis of accuracy for different TTA methods on DEAP with missing data, relative to the baseline method "No Adapt". The No Adapt baseline corresponds to a model pretrained on the source domain and directly evaluated on the target domain without any adaptation. Results report the improvement rate of each TTA method over "No Adapt" at mask ratios of 0.2, 0.4, 0.6, and 0.8, along with the average improvement.

| | Valence(%) | | | | | Arousal(%) | | | | |
| --- | --- | --- | --- | --- | --- | --- | --- | --- | --- | --- |
| Mask ratio | 0.2 | 0.4 | 0.6 | 0.8 | Avg | 0.2 | 0.4 | 0.6 | 0.8 | Avg |
| Tent | -0.162 | 0.000 | -0.314 | -0.162 | -0.162 | 0.471 | -0.469 | 0.167 | 0.232 | 0.101 |
| EATA | -0.315 | 0.157 | -0.154 | 0.076 | -0.061 | 0.701 | -0.705 | 0.305 | 0.234 | 0.134 |
| READ | **1.562** | 0.625 | 0.234 | 0.703 | 0.781 | 0.312 | -0.070 | 0.391 | 0.859 | 0.372 |
| 2LTTA | 0.937 | 0.937 | 0.546 | 1.010 | 0.858 | 0.390 | 0.156 | 0.937 | 0.234 | 0.429 |
| **BiM-TTA(ours)** | 1.406 | **1.250** | **0.859** | **1.172** | **1.172** | **1.016** | **1.719** | **1.094** | **1.406** | **1.309** |

Table 3: Comparative analysis of accuracy of different TTA methods on MAHNOB-HCI with missing data, relative to the baseline method No Adapt. Results show the improvement rate of each TTA method over "No Adapt" at the four mask ratios and the average improvement.

| | Valence(%) | | | | | Arousal(%) | | | | |
|---|---|---|---|---|---|---|---|---|---|---|
| Mask ratio | 0.2 | 0.4 | 0.6 | 0.8 | Avg | 0.2 | 0.4 | 0.6 | 0.8 | Avg |
| Tent | -0.185 | 0.120 | -0.291 | 0.183 | -0.043 | 0.529 | 0.046 | -0.370 | 0.139 | 0.086 |
| EATA | 0.556 | -0.185 | 0.046 | 0.265 | 0.171 | 0.635 | 0.185 | -0.139 | **0.185** | 0.217 |
| READ | 0.523 | 0.370 | 0.079 | -0.741 | 0.058 | 0.529 | -0.741 | -0.556 | 0.079 | -0.146 |
| 2LTTA | 0.450 | -0.218 | 0.079 | 0.185 | 0.124 | 0.575 | -0.324 | -0.185 | 0.000 | 0.017 |
| **BiM-TTA(ours)** | **2.413** | **0.787** | **0.787** | **0.370** | **1.089** | **0.866** | **0.417** | **0.556** | **0.185** | **0.506** |

For experiments with missing data, Table 2 and Table 3 present the performance improvements of BiM-TTA and other TTA methods relative to the "No Adapt" method. "No Adapt" denotes a model trained on the source domain and subsequently evaluated on the target domain containing missing data, without any adaptation. Tent improves model performance on the target domain by minimizing predictive entropy. READ fine-tunes the self-attention fusion layer to adjust inter-modal weights, alleviating the impact of information disparity across different modalities. 2LTTA employs a two-level objective function that includes entropy-based sample reweighting and diversity-promoting loss. However, they indiscriminately adapt the model to all samples with missing data at once, which may result in significant parameter adjustments to fit the target domain, thereby reducing overall performance. Moreover, EATA selectively adapts to reliable samples. However, it does not account for the varying levels of degradation across different modalities and thus does not fully integrate the useful complementary information between modalities. In contrast, our method uses two-level entropy-based sample filtering to avoid the model directly adapting to samples with strong distribution shifts and limited multimodal information. Furthermore, we utilize inter-modal information sharing to align the information between different modalities, facilitating the full adaptation of the model.

In summary, compared to all baselines, BiM-TTA achieves the best results in both cases for two datasets. To further support these results, we provide additional analyzes in the appendices: **Appendix A.7** reports ablation and component analyzes, **Appendix A.8** presents visualization experiments that illustrate the key challenges addressed by our method, **Appendix A.9** presents hyperparameter studies, and **Appendix A.10** evaluates computational efficiency.

## 5  Conclusion

In this paper, we propose a multimodal BiMamba network with TTA for emotion recognition. In the training phase, the multimodal BiMamba network effectively captures intra-modal dependencies and inter-modal correlations of multimodal physiological signals. The two-level entropy-based sample filtering and mutual information sharing across modalities achieve smooth adaptation to the target domain and reduce distribution shifts across modalities, thereby alleviating the negative impact of amplified distribution shifts caused by missing multimodal data. Experiment results demonstrate that our model achieves state-of-the-art performance. Moreover, the ablation studies further confirm the contribution of each component within BiM-TTA. We also discuss the limitations of BiM-TTA in the Appendix A.11. In summary, we design a general multimodal backbone that incorporates a multimodal TTA mechanism. We will further extend BiM-TTA to broader physiological analysis tasks, including sleep stage classification and motor imagery.

## Acknowledgments

This work is supported by the Youth Science Fund Project of National Natural Science Foundation of China (No.62306317), and sponsored by Beijing Nova Program (Grant No. 20250484804).

## Contribution Statement

Tingyu Du, Zhengyu Tian, and Hongkai Li have equal contributions to this paper.

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

# Appendix

## A.1 Further Discussion of Related Work

### A.1.1 Mamba

Structured state space sequence models (SSMs)[44] have emerged as a promising class of architectures for sequence modeling. For example, Gu et al. [45] developed the Structured State Space Sequence Model (S4), which significantly improves computational efficiency by combining the HIPPO matrix with efficient computation methods. Building upon S4, Gu and Dao [46] proposed Mamba, which designs a dynamic selection mechanism to filter out irrelevant information and extract global features more effectively. Mamba also introduces a hardware-aware algorithm for efficient computation, showing great potential in long-sequence modeling. Mamba has demonstrated excellent performance across various tasks. Following Mamba, Xu et al. [47] proposed SST, a model that integrates the strengths of Transformer and Mamba to capture both global and local sequence dependencies for general time-series prediction tasks. In the emotion recognition task using physiological signals, Gui et al. [37] proposed EEG-Mamba, which utilizes BiMamba to better encode EEG physiological signals. It also introduces a task-aware model equipped with general experts to learn task-specific representations from EEG data.

### A.1.2 Test-Time Adaptation

Test-time adaptation (TTA) focuses on adapting source models to target domains without access to source domain data or target domain labels. Traditional domain adaptation requires training with both source and target domain data. In contrast, test-time training[48] enhances test-time adaptability by training source models with supervised and self-supervised objectives in the training stage. However, these methods rely on proxy tasks and assume the training process is accessible, which limits their application scope. TTA addresses this by adjusting models only during testing without intervening in the training stage. For instance, Wang et al. [49] proposed Tent, which updates model normalization layers by minimizing entropy during test. Niu et al. [40] introduced EATA, employing a sample selection criterion based on entropy minimization. Recently, Yang et al. [25] developed READ, which addresses reliability bias by modulating the attention-based fusion layers in a self-adaptive manner and designing a novel objective function for robust multimodal adaptation. Lei and Pernkopf [26] proposed 2LTTA, using Shannon entropy as the objective for the Transformer encoder of the corrupted modality and a diversity-promoting loss as objective for the modality fusion block.

## A.2 Selection of Entropy Threshold

**Entropy Threshold Design**    We adopt a soft-thresholding mechanism in which the entropy threshold is adaptively adjusted for each batch. The threshold is defined as the batch entropy expectation plus or minus n times its variance. By Chebyshev's inequality, this formulation provides a probabilistic bound on sample retention while mitigating the effect of outliers. Empirical results show that setting $n = 2$ achieves optimal performance, with at least 80% of samples retained across both datasets, ensuring effective utilization of test data. To further improve flexibility, we introduce a smoothing factor that gradually relaxes the threshold during inference until it reaches the precomputed batch-adaptive value, allowing the model to incorporate more representative test samples.

**Impact of Threshold Strictness**    Both overly strict and overly loose thresholds degrade performance, and thus an appropriate threshold needs to be determined empirically. When the threshold is too strict, only a small number of samples are retained, which restricts the model's ability to learn the target-domain distribution. In contrast, when the threshold is too loose, many low-quality samples are included, increasing the risk of adapting to low-quality data and compromising model stability.

## A.3 Algorithm for Multimodal TTA

---

**Algorithm 1** Multimodal TTA

---

**Input**: Target samples X.

1: Initialize $X_{\text{remain}} = X$ and $S(x) = \varnothing$.
2: **while** $t = 1$ to $iter$ and $X_{\text{remain}} \neq \varnothing$ **do**
3:   Calculate entropy of multimodal outputs and unimodal outputs for $X_{\text{remain}}$ using Eq.11, and Eq.12.
4:   Calculate multimodal and unimodal entropy bounds using Eq.13, and Eq.14.
5:   Select $S(x)$ from $X_{\text{remain}}$ based on entropy criteria using Eq.15.
6:   Calculate mutual information sharing loss $\mathcal{L}_{\text{mis}}(x)$ for $S(x)$ using Eq.17.
7:   Calculate total loss $\mathcal{L}_{\text{test}}$ for $S(x)$ using Eq.19.
8:   Update tunable parameters of the model using $\mathcal{L}_{\text{test}}$.
9:   Update samples: $X_{\text{remain}} = X_{\text{remain}} - S(x)$.
10: **end while**
11: **return** Adapted prediction probabilities and loss values.

---

## A.4 Description of Datasets

*1) The DEAP dataset:* It involves 32 participants, each completing 40 one-minute trials based on music videos. In the experiment, EEG is recorded with a BioSemi 32-channel cap following the international 10–20 system, while 8 additional channels capture peripheral physiological signals including respiration rate, EOG, and EMG. After each video, participants provide self-assessments of arousal, valence, liking, dominance, and familiarity. During preprocessing, all signals are resampled to 128 Hz. The EEG is bandpass filtered between 4 and 45 Hz, and ocular artifacts are removed. Arousal and valence are rated on a scale from 1 to 9. A threshold of 5 is used to distinguish low and high emotional classes.

*2) The MAHNOB-HCI dataset:* It includes 30 subjects, each completing 20 video-based trials with self-reported ratings from 1 to 9. In the experiment, EEG signals are recorded using the BioSemi Active II system equipped with 32 Ag/AgCl electrodes, while six peripheral physiological signals are simultaneously recorded. These peripheral physiological signals include ECG, GSR, respiration, and body temperature. To maintain consistency across modalities, all recordings are kept at their original sampling rate of 256 Hz. A fixed threshold of 5 is applied to discretize the continuous arousal and valence ratings into binary categories. As the recordings of subjects 12, 15, and 26 are unavailable, the final analysis is conducted on 27 subjects, consistent with previous studies such as TSception and HetEmotionNet[14, 9].

## A.5 The Other Hyperparameters Settings

Table 4: The values of the hyperparameters described in the paper

| Parameter | DEAP | MAHNOB-HCI |
|---|---|---|
| The auxiliary task training weights $\alpha_i$ | [0.8, 0.3, 0.2] | [0.5, 0.05, 0.03] |
| The TTA parameter $\lambda$ | 1 | 0.3 |
| The unimodal entropy weights $\mu_i$ | [1, 0.3, 0.2] | |
| The total number of iterations $iter$ | 7 | |
| The TTA parameter $\beta$ | 0.2 | |
| The learning rate of TTA | 0.0007 | |

## A.6 Detailed Introduction to the Baseline Methods

We compare our model approach with ten state-of-the-art emotion recognition models in the same domain, including:

- SVM[50]: A traditional machine learning method that utilizes statistical features, wavelet-based energy-entropy, RMS, and other techniques to construct feature vectors for emotion recognition.
- EEGNet[13]: A lightweight convolutional neural network designed for EEG-based brain–computer interface applications.
- ACRNN[51]: A model that combines a channel-wise attention mechanism, a convolutional neural network (CNN), a recurrent neural network (RNN), and an extended self-attention mechanism.
- HetEmotionNet[9]: A model that fuses multimodal physiological signals for emotion recognition using a two-stream heterogeneous graph recurrent neural network, capturing spatial-spectral-temporal features, heterogeneity, correlation, and dependencies.
- TSception[14]: A multi-scale convolutional neural network combining dynamic temporal layer, asymmetric spatial layer, and high-level fusion layer.
- LGGNet[52]: A model that models local-global-graph representations of EEG through multiscale temporal convolutions and local- and global-graph-filtering layers.
- EEG-Deformer[53]: A model that incorporates a Hierarchical Coarse-to-Fine Transformer (HCT) block and a Dense Information Purification (DIP) module into a CNN-Transformer.
- MambaFormer[47]: A hybrid sequence model that embeds Mamba modules into Transformer feed-forward layers, combining state-space efficiency with attention expressiveness for modeling both long- and short-range temporal dependencies.
- SST[47]: A hybrid model that couples a Mamba-based global expert with a windowed Transformer local expert, fusing their outputs through a routing mechanism to strengthen long-range temporal modeling.
- VSGT[6]: A graph-based model for EEG emotion recognition that leverages a variational spatial encoder and a Gaussian temporal encoder to incorporate prior knowledge and model spatial and cross-temporal dependencies across brain regions.

We then compare our TTA method against the following baseline methods:

- No Adapt: A baseline that evaluates a model pretrained on source domain data directly on target domain data without performing adaptation.
- Tent[49]: A method that adapts models at test time by minimizing prediction entropy through updating batch normalization parameters.
- EATA[40]: A method that filters reliable samples based on prediction entropy and assigns adaptive weights to both samples and important parameters.
- READ[25]: A multimodal TTA method that mitigates reliability bias by employing a new paradigm that modulates the attention between modalities in a self-adaptive way and adopting a novel objective function for robust multimodal adaptation.
- 2LTTA[26]: A multimodal TTA method that addresses intra-modal distribution shifts and cross-modal reliability bias in multimodal learning by modulating normalization layers and self-attention modules and employing a two-level objective function with entropy-based sample reweighting and diversity-promoting loss.

### A.7 Ablation Studies and Component Analysis

To examine the overall contribution of major components, we first conduct ablation experiments on the multimodal BiMamba network and multimodal TTA. As shown in Figure 2, the results indicate that the removal of both the multimodal BiMamba network and multimodal TTA leads to a decrease in model performance, with the absence of multimodal TTA causing a more significant performance drop. This demonstrates that multimodal TTA is more effective on datasets with missing multimodal data. Overall, both the multimodal BiMamba network and multimodal TTA are effective for emotion recognition in the proposed model.

Beyond this coarse-level analysis, we further design fine-grained experiments to evaluate the effectiveness of subcomponents: (1) comparative experiments for the two BiMamba modules without missing multimodal data, and (2) ablation studies of Multimodal TTA with missing multimodal data.

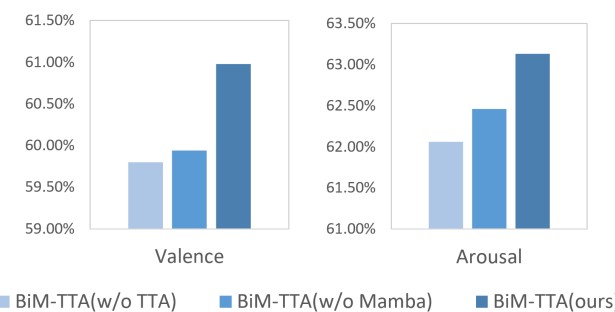

Figure 2: Ablation study on DEAP with missing data. "w/o Mamba" indicates the absence of the BiMamba in our network, and "w/o TTA" indicates the absence of the multimodal TTA. The results report the average performance across the four mask ratios.

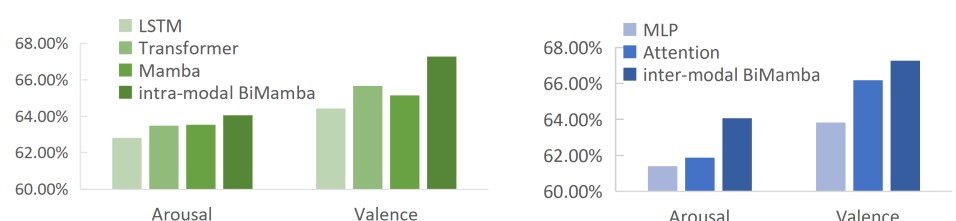

Figure 3: Comparison of intra-modal BiMamba and inter-modal BiMamba with other methods

For the intra-modal BiMamba module, we compare its performance against Mamba, Transformer, and LSTM architectures to evaluate its intra-modal sequence modeling capabilities. For the inter-modal BiMamba module, we compare it with attention-based fusion and MLP fusion methods to assess its ability to integrate multimodal features. The corresponding experimental results are shown in Figure 3, which demonstrate that both intra- and inter-modal BiMamba consistently outperform the alternatives, highlighting the superior modeling capacity of our BiMamba modules.

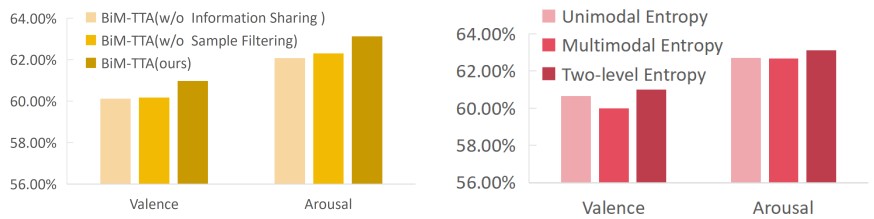

Figure 4: Ablation study on TTA and Two-level Entropy-based Sample Filtering

For the multimodal TTA, we perform ablation studies from two perspectives: the impact of removing individual components and the effectiveness of single-level entropy-based filtering. The results are summarized in Figure 4, demonstrating that each component has a beneficial impact on the overall performance.

## A.8 Visualization Experiments

To better illustrate how BiM-TTA addresses the two key challenges of multimodal emotion recognition, we provide two types of visualization experiments.

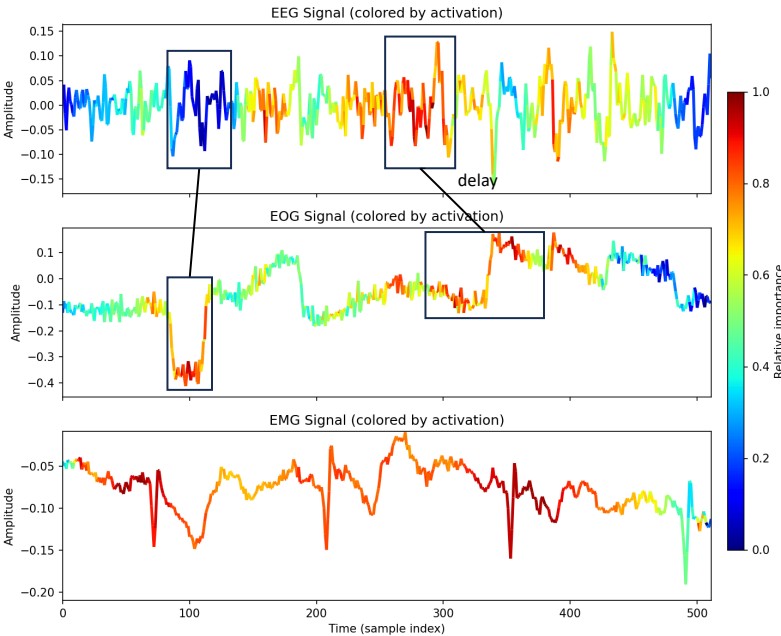

Figure 5: Grad-CAM visualization of BiM-TTA

First, Figure 5 shows Grad-CAM visualizations of BiM-TTA, which illustrate how the model captures intra-modal long-range dependencies as well as inter-modal correlations. The EMG signal demonstrates that the model effectively captures long-range intra-modal features. Moreover, the following inter-modal correlations emerge: 1) when the EOG shows pronounced activation between time steps 80 and 120, the resulting high-frequency motion artifacts (ocular artifacts) contaminate the EEG signal[54], and the model mitigates this interference by reducing the attention weights assigned to those EEG segments. 2) activation in the EEG between time steps 260 and 300 provokes a delayed eye-movement response in the EOG, which appears between time steps 290 and 350[55]. These dynamics align closely with the physiological patterns documented in emotion-recognition research.

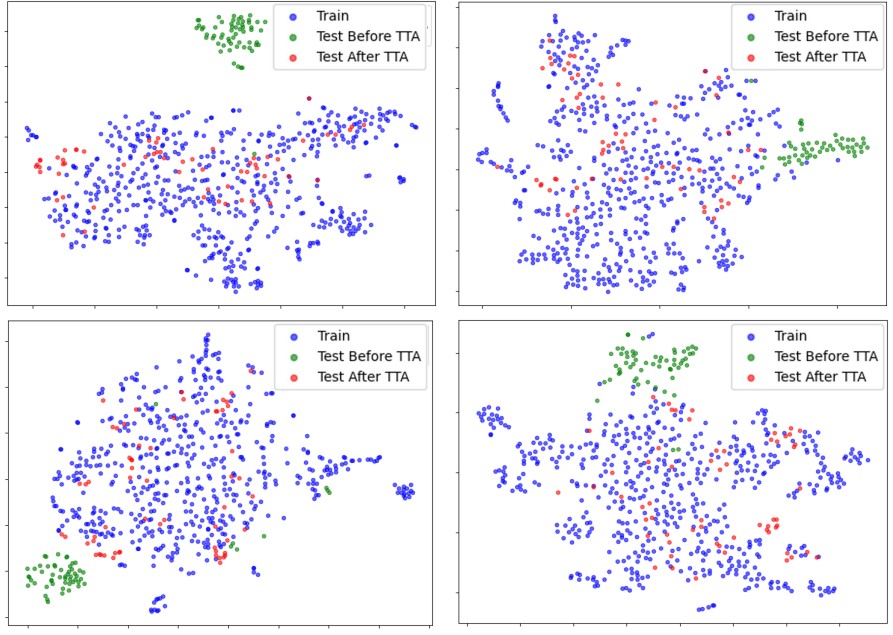

Figure 6: t-SNE visualization of TTA

Second, Figure 6 presents a t-SNE visualization of feature distributions, clearly demonstrating that BiM-TTA can mitigate the distribution shifts induced by missing multimodal data. The green points represent the feature distribution extracted by the model from the original test set before applying TTA, while the blue points correspond to the training distribution. After applying TTA, the test distribution progressively aligns with the training distribution, eventually appearing as the red points. This progressive alignment demonstrates that TTA is effective in mitigating the distribution gap between the test and training sets, thereby enabling the model to perform emotion recognition more effectively under testing conditions.

## A.9 Hyperparameter Studies

To further analyze the effect of hyperparameter settings, we examine five key hyperparameters: the loss weight of unimodal auxiliary tasks $\alpha_i$, the loss-balance coefficient $\lambda$, the weight of unimodal entropy $\mu_i$, the total number of iterations $iter$, and the initial value of the smoothing factor $\beta$.

Table 5: Studies on the Auxiliary task weights of multimodal BiMamba on DEAP with missing data. The weights correspond to the parameters for EEG, EOG, and EMG. The results report the average performance across the four mask ratios.

|      |     | Valence | Arousal |
|------|-----|---------|---------|
|      | 0.7 | 0.608   | 0.627   |
| EEG  | 0.8 | **0.610** | **0.631** |
|      | 0.9 | 0.609   | 0.624   |
|      | 0.2 | 0.609   | 0.626   |
| EOG  | 0.3 | **0.610** | **0.631** |
|      | 0.4 | 0.607   | 0.625   |
|      | 0.1 | 0.604   | 0.623   |
| EMG  | 0.2 | **0.610** | **0.631** |
|      | 0.3 | 0.605   | 0.620   |

The results for the auxiliary-task weight $\alpha_i$ are reported in Table 5. Based on experimental observations, the EEG branch plays a more critical role than EOG and EMG in classification performance. Therefore, its auxiliary task is assigned a larger weight.

The effects of the remaining four hyperparameters ($\lambda$, $\mu_i$, $iter$, and $\beta$) are illustrated in Figure 7. For $\lambda$ and $\mu_i$, the model remains stable across a wide range of values, indicating that performance is not sensitive to fine-tuning these coefficients. For $iter$, performance peaks within a moderate range, where the model adapts effectively without overfitting. Too few iterations lead to insufficient adaptation, while too many cause mild over-adaptation and slightly degrade performance. For $\beta$, relatively small values achieve the best balance between reliability and adaptability. A smaller $\beta$ enforces a stricter initial threshold, retaining only samples with rich multimodal information and high prediction confidence. In contrast, a larger $\beta$ relaxes the threshold too early, potentially admitting low-quality samples and disrupting learned representations, which degrades performance. Overall, although different hyperparameter choices lead to slight variations in performance, the model consistently outperforms the No Adapt setting, indicating that the proposed method remains robust across a broad range of hyperparameter configurations.

To gain deeper insight into the adaptation dynamics, we further analyze the performance trend of TTA during the iterative process, as shown in Figure 8. With the iteration parameter $iter$ set to 7, the model performance progressively improves over iterations and reaches its optimal average level at the final step.

## A.10 Computational Efficiency Analysis

We compare the computational cost required to perform a complete inference for a single subject on the DEAP dataset. To ensure fairness and accuracy, we use the official open-source implementations of the latest best-performing baseline models for comparison. As shown in Table 6, our model is lightweight, maintains real-time performance, and achieves an optimal trade-off between performance and efficiency. All experiments are conducted on an NVIDIA RTX A4000 (16G). A single training run (30 epochs) takes approximately 60 minutes on a single GPU.

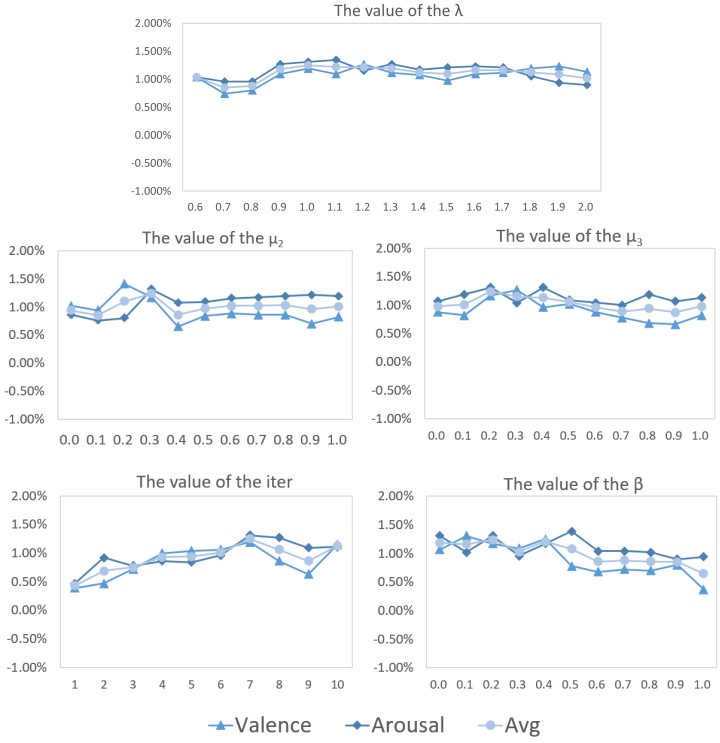

Figure 7: Hyperparameter studies of BiM-TTA on DEAP with missing data. The parameters include $\lambda$, $\mu_2$, $\mu_3$, $iter$, and $\beta$, with $\mu_1$ set to 1 by default. The results report the average improvement across the four mask ratios.

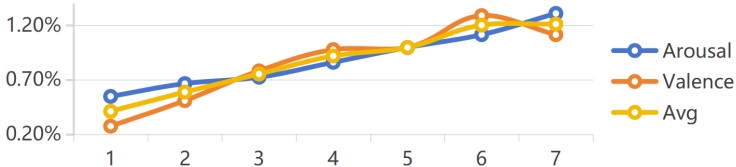

Figure 8: Model Performance across TTA Iterations

Table 6: Comparison of computational overhead parameters with two recent state-of-the-art models

| Model | Parameter Count | Inference Time (s) | FLOPs (M) | GPU Memory (MB) |
|---|---|---|---|---|
| LGGNet | 721,319 | 0.73 | 10.75 | 95.40 |
| EEG-Deformer | 915,394 | **0.43** | 5.39 | 8.21 |
| **BiM-TTA(ours)** | **22,431** | 0.47 | **1.18** | **2.29** |

## A.11 Limitations

We would like to discuss some of the limitations identified during this study. Firstly, although BiM-TTA demonstrates outstanding performance in this study, its latency and resource consumption in real-time online systems (e.g., wearable devices) have not yet been evaluated. In future work, we plan to apply pruning and quantization, perform distillation, or design lightweight variants to meet the demands of real-time embedded scenarios. Secondly, although we have proposed a general model architecture, its applicability in multi-task settings remains insufficiently validated. Going forward, we will assess the effectiveness of BiM-TTA on tasks such as sleep stage classification and motor imagery.

