# OpenReview forum: "A Multimodal BiMamba Network with Test-Time Adaptation for Emotion Recognition Based on Physiological Signals"
_NeurIPS.cc/2025/Conference — NeurIPS 2025 poster_

### Official Review · Reviewer_Z1w5 · 2025-06-21

**Clarity:** 3
**Significance:** 3
**Originality:** 4
**Rating:** 5
**Confidence:** 4

**Summary:**

This paper proposes a multimodal network BiMamba with test time adaptation TTA for emotion recognition. It addresses long-range dependence and inter-modal correlation in multimodal physiological signals, and it mitigates performance degradation of missing data.

**Questions:**

1. For the first questions, please refer to Weaknesses 2.
2. How to define the number of iterations? According to the experimental results in Figure 6 in the appendix, it can still be improved.
3. What is the basis for setting the two thresholds? What is the impact of different thresholds on the model results?

**Ethical Concerns:**

["NO or VERY MINOR ethics concerns only"]

**Limitations:**

yes

**Quality:**

4

**Strengths And Weaknesses:**

Strengths:
1. Introducing TTA to sold missing data problem is a good and innovative method. The reason of using mamba is clear and reasonable.
2. This paper is well written with a clear structure, such as the correspondence between formulas and figures. The experiment section and appendix provide solid details.
3. The problem solved by this paper is meaningful and not often mentioned. Therefore, the motivation is reasonable.

Weaknesses:
1. This paper puts most of the experiment in the appendix, but the guidance is not clear enough.
2. The explanation of two-level entropy filtering is not clear enough. This paper mentions that these two filters seem to use different filter criteria and seem to have opposite threshold selections (one tends to be high and the other tends to be low). The large amount of text in this part makes it more difficult to sort out the relationship between them.
3. Figure 1 needs to be improved. The author seems to want to use color matching to illustrate the correlation between the components. But there are too many colors in the figure, and the colors of components and the background are relatively close and difficult to identify.

---

> ### Author Rebuttal · Authors · 2025-07-30
>
> # Response
>
> ## W1
>
> 　　Thank you for your valuable suggestions. We have optimised the manuscript structure as follows:
>
> 1.  **We have relocated several key experiments from the appendix to the main experimental section.**
> >Considering that certain key experiments, such as ablation studies and hyperparameter analyses, are crucial for validating the model's effectiveness, we have moved these specific experiments from the appendix to the main experimental section. We have also supplemented them with more detailed design descriptions and result analyses to improve the completeness and readability of the main text.
>
> 2.  **We have added an "Experimental Overview" subsection.**
> >To improve the overall readability of the main text, we add a new subsection titled "Experimental Overview" to briefly summarize the design and key findings of each experiment. At the same time, we provide clear guidance directing readers to the more detailed figures and analyses in the appendix, enabling a more efficient understanding of the core contributions.
>
> *Manuscript revision:* We have added subsections "4.4 Experimental Overview", "4.5 Ablation Experiments", and "4.6 Hyperparameter Effects" in the revised manuscript.
>
> ---
>
> ## W2 & Q1
>
> **We have streamlined the description of two-level entropy-based sample filtering strategy and added a schematic diagram to aid intuitive understanding.** This mechanism is designed to prioritize the selection of **samples with high unimodal entropy and low multimodal entropy** for fine-tuning. Specifically, it includes the following two aspects:
>
> >I.  **Unimodal entropy $Ent(x_i)$ measures a sample's reliance on a single modality. Higher values indicate suitability for fine‐tuning.**
> >- *Low $Ent(x_i)$:* Model can predict using one dominant modality, implying limited multimodal information.
> >- *High $Ent(x_i)$:* Model likely integrates multiple modalities for prediction, indicating rich multimodal information.
>
> >II.  **Multimodal entropy $Ent(x)$ reflects prediction certainty. Lower values indicate suitability for fine‐tuning.**
> >- *Low $Ent(x)$:* Predictions are more certain. This helps reduce uncertainty in target domain predictions.
> >- *High $Ent(x)$:* Predictions are uncertain. The uncertainty in predictions may introduce misleading information and compromise model stability, therefore making such samples unsuitable for fine-tuning.
>
> *Manuscript revision:* We have updated "3.3.1 Two-level Entropy-based Sample Filtering" and added the schematic diagram.
>
> ---
>
> ## W3
>
> 　　Thank you for the detailed feedback on Figure 1's colour scheme. We have simplified the colour palette and increased foreground-background contrast to more clearly delineate components and overall structure.
>
> ---
>
> ## Q2
>
> **We determine the number of iterations $iter$ experimentally.**
> >In the experiment labeled "The value of the iter," shown in the lower-left panel of Figure 2, we systematically evaluate performance under various settings and ultimately set $iter=7$. Using this selected value, Figure 6 then illustrates how model performance evolves from iteration 1 through 7, showing steady improvement and peak performance at iteration 7.
>
> ---
>
> ## Q3
>
> 　　Thank you for your question. Below is our response:
>
> 1.  **The entropy threshold is determined jointly by batch‐level statistics and hyperparameter tuning.**
> >Inspired by Chebyshev's inequality, we set the threshold to the batch's mean entropy plus $n$ times its variance, where $n$ is a hyperparameter controlling the strictness of sample filtering. This design aims to retain high‑quality samples while suppressing interference from outlier samples. The detailed methodology for selecting the entropy threshold is provided in "A.3 Selection of Entropy Threshold", and we will emphasize this discussion more prominently in the main text to clarify its role within the overall method.
>
> 2.  **Both overly strict and overly loose thresholds degrade performance, and thus an appropriate threshold needs to be determined through empirical evaluation.**
> >-  *Too strict:* Too few samples pass the filter, limiting learning of the target‐domain distribution.
> >-  *Too loose:* A large number of samples are accepted, including outliers, which increases the risk of adapting to noisy data.

---

> > ### Comment · Reviewer_Z1w5 · 2025-08-06
> >
> > Thank you for the author's reply, which has solved most of my problems. I will maintain my positive score

---

> > > ### Author Response · Authors · 2025-08-08
> > > **Appreciation to Reviewer Z1w5**
> > >
> > > Dear Reviewer,
> > >
> > > ---
> > >
> > > Thank you for your careful review. I'm glad our response resolved most of your concerns and appreciate your continued positive score.
> > >
> > > ---
> > >
> > > Best Regards, Authors of Paper ID 8789

---

### Official Review · Reviewer_RRS6 · 2025-06-26

**Clarity:** 3
**Significance:** 3
**Originality:** 4
**Rating:** 4
**Confidence:** 4

**Summary:**

The paper presents BiM-TTA for emotion recognition. The contribution is to handle temporal dependencies, modal correlations, and missing data problems.

**Questions:**

1)Why is there no consideration of the case where a modality is completely missing?
2)The contribution and importance of different modality do not seem to be explicitly considered?
3)How is the Entropy threshold selected?
4)What is the impact of different initial values or smoothing factors on the model?
5)This paper emphasizes that Mamba is an architecture that is good at solving long-distance dependencies, so why is Mamba still used when learning the relationship between modalities? The number of modalities in this paper is not large.

**Ethical Concerns:**

["NO or VERY MINOR ethics concerns only"]

**Final Justification:**

The authors have addressed most of my concerns. The proposed method shows novelty, and the performance on two datasets is promising. Therefore, I have given a positive score.

**Limitations:**

YES

**Quality:**

3

**Strengths And Weaknesses:**

# Strengths:
- The motivation is clear and reasonable.
- The entropy-based TTA is novel and mitigates missing data issues.
- Performance on two datasets is good. Also this paper provides a lot of extra experiments.
# Weaknesses:
- The structure of the paper is not appropriate. A large number of experiment are placed in the appendix, and although many of them provide important impacts, these are not mentioned and explained in the main text.
- There is insufficient explanation of some details, such as the basis for the selection of some hyperparameters. The main text does not summarize some the hyperparameter experiments, making it difficult to establish a connection between the supplementary experiments in the appendix and the arguments in the main text.

---

> ### Author Rebuttal · Authors · 2025-07-30
>
> # Response
>
> ## W1
>
> 　　Thank you for your valuable suggestions. We have optimized the manuscript structure as follows:
>
> 1.  **We have relocated several key experiments from the appendix to the main experimental section.**
>
>     >Considering that certain key experiments, such as ablation studies and hyperparameter analyses, are crucial for validating the model's effectiveness, we have moved these specific experiments from the appendix to the main experimental section. We have also supplemented them with more detailed design descriptions and result analyses to improve the completeness and readability of the main text.
>
> 2.  **We have added an "Experimental Overview" subsection.**
>
>     >To improve the overall readability of the main text, we add a new subsection titled "Experimental Overview" to briefly summarize the design and key findings of each experiment. At the same time, we provide clear guidance directing readers to the more detailed figures and analyses in the appendix, enabling a more efficient understanding of the core contributions.
>
> *Manuscript revision:* We have added subsections "4.4 Experimental Overview", "4.5 Ablation Experiments", and "4.6 Hyperparameters Effection" in the revised manuscript.
>
> ---
>
> ## W2
>
> 　　Thank you for your valuable feedback regarding the basis for the selection of some hyperparameters and the summary of experiments. We have added relevant content in the main text, as outlined below:
>
> 1.  **Explanation of hyperparameter selection.**
>
>     >In "4.6 Hyperparameter Effects" of the main text, We have supplemented the basis for choosing key hyperparameters, such as the entropy threshold and smoothing factor, and analyzed their performance curves under varying values. The basis for selecting the entropy threshold and the smoothing factor can be found in our responses to Q3 and Q4.
>
> 2.  **Summary and connection of hyperparameter experiments.**
>
>     >We have added a new subsection "4.4 Experimental Overview," which briefly summarizes the impact of each hyperparameter on model performance and directs readers to the corresponding experimental details in the manuscript.
>
> *Manuscript revision:* We have added subsections "4.4 Experimental Overview", "4.5 Ablation Experiments", and "4.6 Hyperparameters Effection" in the revised manuscript.
>
> ---
>
> ## Q1
>
> **The case where a modality is completely missing rarely occurs in physiological signal acquisition experiments.**
> >In actual physiological signal collection, complete modality missing is usually detected in time and fixed by inspecting or replacing the device. Therefore, our study mainly focuses on the more common problem of partial modality data missing.
>
> ---
>
> ## Q2
>
> **We regulate the contribution of different modalities in the model by adjusting the loss weight of the auxiliary task and the hyperparameter of Mutual Information Sharing Across Modalities.**
> >These parameters reflect the relative importance and influence of each modality. To further enhance interpretability, we plan to add a weight-visualization analysis for the inter-modal BiMamba in a future version, which will intuitively illustrate each modality's specific contribution.
>
> ---
>
> ## Q3
>
> **The entropy threshold is determined jointly by batch-level statistics and hyperparameter tuning.**
> >Inspired by Chebyshev's inequality, we set the threshold to the batch's mean entropy plus $n$ times its variance, where $n$ is a hyperparameter controlling the strictness of sample filtering. This design aims to retain high‑quality samples while suppressing interference from outlier samples. The detailed methodology for selecting the entropy threshold is provided in "A.3 Selection of Entropy Threshold", and we will emphasize this discussion more prominently in the main text to clarify its role within the overall method.
>
> ---
>
> ## Q4
>
> **The initial value $\beta$ and the smoothing factor $\beta_t$ influence both the number and quality of samples entering the adaptation process at each iteration.** The smoothing factor $\beta_t$ is jointly determined by the initial value $\beta$ and the growth strategy, with the specific effects described as follows:
>
> >I.  **Initial $\beta$:** A smaller $\beta$ yields a stricter initial threshold, retaining only samples with rich multimodal information and high prediction confidence, thereby enhancing stability at early testing phase. A larger $\beta$ relaxes the threshold, potentially admitting low‐quality samples too soon and disrupting learned representations, which may degrade performance.
> >
> The optimal value of $\beta$ is determined experimentally and set to 0.2, with relevant results shown at "The value of the $\beta$" in the lower right corner of Figure 2.
>
> >II.  **Growth strategy:** After determining a suitable $\beta$, we adopt the following growth strategy:
>     $$\beta_t = \beta + (1 - \beta)\cdot\frac{t}{\mathrm{iter}}$$
>     where $\beta_t$ increases linearly with iterations, gradually relaxing the entropy threshold. This process allows more samples to be sequentially incorporated into the adaptation process and enables the model to more effectively cover the target domain.
>
> In summary, by setting an appropriate initial $\beta$ and growth rate, this strategy balances early robustness with later adaptability, facilitating a smooth, high-quality model adaptation.
>
> ---
>
> ## Q5
> **In our paper, we emphasize both Mamba’s strength in modeling long‑range dependencies and its ability to capture inter‑modal correlations.**
> >Specifically, besides effectively capturing long-range temporal dependencies, its explicit global state variables, as part of the state space model (SSM), also effectively characterize correlations between different modalities. Based on these considerations, we ultimately select Mamba as the core component of the backbone network. As briefly mentioned in line 82 of the introduction, this topic is initially discussed. In subsequent versions, we will further elaborate to aid readers' deeper understanding.

---

> > ### Comment · Reviewer_RRS6 · 2025-08-04
> >
> > Thanks for the clarifications. The authors have solved most of my confusion and questions. I will maintain my postive score.

---

> > > ### Author Response · Authors · 2025-08-08
> > > **Appreciation to Reviewer RRS6**
> > >
> > > Dear Reviewer,
> > >
> > > ---
> > >
> > > Thank you for taking the time to review our revisions. We're glad most concerns were resolved and truly appreciate your positive score.
> > >
> > > ---
> > >
> > > Best Regards, Authors of Paper ID 8789

---

### Official Review · Reviewer_TeXe · 2025-06-30

**Clarity:** 3
**Significance:** 3
**Originality:** 2
**Rating:** 4
**Confidence:** 4

**Summary:**

This paper proposes BiM-TTA, a multimodal emotion recognition method based on physiological signals that integrates a bidirectional Mamba (BiMamba) network with a dedicated test-time adaptation (TTA) strategy. The BiMamba network consists of two modules: (1) an intra-modal BiMamba that models long-range temporal dependencies within each physiological modality and (2) an inter-modal Bimodal Bi-Mamba which captures cross-modal correlations by operating along the channel dimension. To address the challenge of missing or corrupted multimodal data at test time, the authors introduce a two-level entropy-based sample filtering mechanism combined with mutual information sharing across modalities during TTA. This adaptation selectively fine-tunes a small subset of model parameters on target domain data, mitigating distribution shifts caused by missing signals.

The method is evaluated on two multimodal emotion corpora (DEAP and MAHNOB-HCI). It is tested under both complete and missing data scenarios, performing better than several strong baselines and recent TTA methods.

**Questions:**

Please clarify why no comparison was made with recent hybrid models.

**Ethical Concerns:**

["NO or VERY MINOR ethics concerns only"]

**Limitations:**

The authors briefly mention limitations in Appendix A.13, acknowledging that BiM-TTA has not yet been validated on tasks beyond emotion recognition, such as sleep stage classification or motor imagery, and that future work is planned in this direction.

**Paper Formatting Concerns:**

Several important experimental details are relegated to the appendix.

**Quality:**

3

**Strengths And Weaknesses:**

Strengths:
1) The paper addresses two substantial and well-motivated challenges in multimodal emotion recognition from physiological signals: (1) modeling long-range intra-modal dependencies and inter-modal correlations, and (2) handling distribution shifts and missing data at test time.
2) The proposed BiMamba architecture is a technically sound and well-justified application of structured state space models (SSMs) in a multimodal context.
3) The test-time adaptation (TTA) mechanism is designed, combining two-level entropy-based sample filtering and mutual information sharing across modalities.
4) The empirical evaluation is thorough, with comparisons against strong baselines on two public corpora  and both complete and missing data conditions.
5) The ablation studies, iterative TTA analysis, Grad-CAM, and t-SNE visualizations convincingly validate the technical components and adaptation process.

Weaknesses:
1) While the TTA mechanism has been engineered, its conceptual novelty is moderate, as there are similar entropy-based selective adaptation strategies. The proposed two-level filtering and gradual threshold relaxation are incremental, rather than fundamentally new.
2) A lack of comparison with Mamba-Transformer hybrid baselines or recent Transformer-based SSM models creates a gap in benchmarking against the closest architectural alternatives.

---

> ### Author Rebuttal · Authors · 2025-07-30
>
> # Response
>
> ## W1
>
> 　　Thank you very much for your attention to and detailed review of our conceptual novelty.
> In response to your questions, we are here to further clarify and supplement two aspects to more clearly demonstrate our innovations in the two-level entropy-based sample filtering strategy and its synergy with the overall architecture:
>
> 1.  **Our work represents an extension and expansion of TTA strategy within the field.**
>
>     >Although entropy-based selective adaptation strategies have been extensively validated in computer vision and related domains, the specific downstream task of “multimodal physiological signals + emotion recognition + data missing” remains largely unexplored. Compared to existing EEG‐based TTA methods such as ATTA[1] and OTTA[2], our method uses entropy in a more distinctive and targeted manner. ATTA updates the adaptive model via unimodal cross-entropy, and OTTA minimizes only classification entropy. In contrast, we introduce both unimodal entropy and multimodal entropy, where the former captures the richness of emotional information present in each sample’s multimodal physiological signals and the latter gauges the model’s confidence in predicting the emotional state. Based on these, we construct a soft-threshold filtering strategy that dynamically removes samples whose emotional features degrade due to missing key information, more finely distinguishing sample quality across emotions. This ensures stability during adaptation while more effectively improving cross‐domain emotion recognition accuracy.
>
> 2.  **The filtering mechanism does not function in isolation. Instead, it serves as an integral part of the overall architecture,working in concert with other modules to jointly enhance the model's adaptive capability during the testing phase.**
>
>     >I. **Synergistic Effect between Two-level Entropy-based Sample Filtering and Mutual Information Sharing Across Modalities.** Our multimodal TTA method comprises two complementary components: a global two-level entropy-based sample filtering strategy and a local cross modalities strategy.
>     >- The global strategy uses a soft threshold jointly constructed from unimodal and multimodal entropy to select a subset of information‐rich, high‐confidence samples, thereby ensuring data quality during adaptation.
>     >- The local strategy, based on these reliable samples, further promotes cross‐modal alignment by minimizing the KL divergence among prediction distributions of each modality.
>     Together, they achieve progressive optimization from global sample filtering to local modal alignment.
>
>     >II. **Synergistic Effect between the Multimodal BiMamba Backbone and TTA Method with Two-level Entropy-based Sample Filtering and Mutual Information Sharing Across Modalities.**
>     >
>     >**The multimodal BiMamba backbone network** first captures emotional features along temporal and channel dimensions, mining long-range temporal dependencies and cross-modal correlations to generate strong base representations.Building on these high-quality features, **the TTA method** applies the two-level entropy-based sample filtering and mutual information sharing across modalities: the former ensures adaptation only on reliable samples, and the latter dynamically adjusts inter-modal information flow within each sample, mitigating interference from imbalanced modality information. The network and TTA collaborate across feature construction and test‐time adaptation to enhance emotion recognition performance.
>
> ---
>
> ## W2 & Q1
>
> 　　As no existing work has explored Mamba-Transformer hybrids in physiological-signal emotion recognition, this direction remains unexplored and was not included in our original comparisons. Following your suggestion, we have added three recent SSM-Transformer approaches for time-series modelling as supplementary baselines, including MambaFormer and SST as Mamba-Transformer hybrid models and Heracles as a Transformer-based SSM method. And we have incorporated the results and analysis of these baselines into "4.3 Experiment Analysis." Thank you for your valuable suggestion.
>
>
>
> *Table: Comparative analysis of accuracy (Valence, Arousal) for different emotion baselines on DEAP and MAHNOB-HCI datasets. * indicates p<0.05.*
>
> | Method             | DEAP          |              | MAHNOB-HCI     |               |
> |--------------------|---------------|--------------|----------------|----------------|
> |                    | Valence       | Arousal      | Valence        | Arousal        |
> | MambaFormer [3]    | 0.621         | 0.587*       | 0.588*         | 0.619          |
> | SST [3]            | 0.613*        | 0.623        | 0.606*         | 0.616*         |
> | Heracles [4]       | 0.633         | 0.611*       | 0.621          | 0.603*         |
> | **BiM‑TTA (ours)** | **0.673**     | **0.641**    | **0.650**      | **0.635**      |
>
>
>
> **Experimental analysis:**
>  MambaFormer embeds the Mamba module into each Transformer encoder’s feed‑forward layer to create a unified long‑and‑short‑range architecture. SST introduces a Mamba‑based global pattern expert alongside a windowed Transformer local variation expert and dynamically fuses their outputs via a long‑short‑range router. Heracles leverages a real‑valued Hartley‑kernel SSM to capture global sequence dependencies, a localized convolutional network to extract fine‑grained details and attention mechanisms in its deeper layers to enable rich temporal token interactions. These models exhibit strong long‑short‑range modeling capabilities. However, their designs focus solely on temporal dynamics and do not explicitly model cross‑channel interactions, which prevents them from capturing inter‑modal relationships and limits their effectiveness in multimodal emotion recognition tasks. Moreover, the complexity of these hybrid architectures can introduce a greater risk of overfitting on relatively small physiological‑signal datasets, further affecting their performance.
>
> *Manuscript revision:* We have updated the results in Table 1 and revised Section 4.3 "Experiment Analysis" accordingly.
>
> ---
> ---
>
> # References
>
> [1] Jia Z, Yang X, Zhou C, et al.  *Atta: Adaptive Test-Time Adaptation for Multi-Modal Sleep Stage Classification*.  Proceedings of the International Joint Conference on Artificial Intelligence (IJCAI), 2024.
>
> [2] Guo H, Li C, Peng H, Han Z, Chen X.  *Test Time Adaptation for Cross-Domain Sleep Stage Classification*.  Biomedical Signal Processing and Control, 2025.
>
> [3] Xu X, Liang Y, Huang B, Lan Z, Shu K.  *Integrating Mamba and Transformer for Long-Short Range Time Series Forecasting*.  CoRR, 2024.
>
> [4] Badri N. Patro, Suhas Ranganath, Vinay P. Namboodiri, Vijay S. Agneeswaran. *Heracles: A Hybrid SSM-Transformer Model for High-Resolution Image and Time-Series Analysis*. arXiv, 2024.

---

> > ### Author Response · Authors · 2025-08-07
> > **Follow-up to Reviewer TeXe**
> >
> > Dear Reviewer,
> >
> > ---
> >
> > As the discussion phase is drawing to a close with less than two days remaining, we would greatly appreciate it if you could kindly let us know whether our responses and the additional experiments have sufficiently addressed your concerns. We would be more than happy to further elaborate within the remaining discussion period.
> >
> > Thank you again for your valuable time and thoughtful engagement throughout the review process.
> >
> > ---
> >
> > Best Regards, Authors of Paper ID 8789

---

### Official Review · Reviewer_krBP · 2025-07-01

**Clarity:** 3
**Significance:** 2
**Originality:** 2
**Rating:** 4
**Confidence:** 2

**Summary:**

In this paper, the authors propose a multimodal bidirectional Mamba (BiMamba) network with test-time adaptation (TTA) for emotion recognition, named BiM-TTA. Specifically, BiM-TTA consists of a multimodal BiMamba network and a multimodal TTA module. The multimodal BiMamba network includes intra-modal and inter-modal BiMamba modules, which model long-range dependencies along the time dimension and capture cross-modal correlations along the channel dimension, respectively. The TTA module effectively mitigates the negative impact of distribution shifts amplified by missing multimodal data through two-level entropy-based sample filtering and mutual information sharing across modalities.

**Questions:**

We believe it is of great need to give the thoughts of the BiMamba design

**Ethical Concerns:**

["NO or VERY MINOR ethics concerns only"]

**Final Justification:**

We have recieved the feedback from the authors. After considering the rebuttal and discussions with authors, other reviewers and AC, we agree to upgrade the rating score from 3 to 4.

**Limitations:**

Yes

**Quality:**

3

**Strengths And Weaknesses:**

The paper is well written and easy to read. And the effectiveness of the proposed method has been validated by the experiments.

The BiMamba seems the one of the main contributions of this work. However, we only see the description of the data processing flow in Bi Mamba instead of the intuition behind the module. Thus, we believe it is of great need to give the thoughts of the BiMamba design.

---

> ### Author Rebuttal · Authors · 2025-07-30
>
> # Response
> ## W1&Q1:
>
> 　　Thank you for your suggestions on the methodology section of our paper. To clarify the design thoughts of the BiMamba module, we have added the corresponding explanation in "3.2.1 Intra-modal BiMamba Module" and "3.2.2 Inter-modal BiMamba Module" of the main text. The module's workflow comprises the following three steps: **(1) Gating mechanism $\to$ (2) Bidirectional state-space modelling $\to$ (3) Linear projection and residual connection.** Specifically, the intuition behind them is:
>
>
> 1.  **Gating mechanism: This mechanism assigns learnable weights to each channel or time step, adaptively emphasizing emotion-relevant representations while suppressing irrelevant information.**
>
>     >In particular, physiological signals not only contain key features such as abrupt emotional change points and dynamic regions that faithfully reflect emotional fluctuations, but also often include environmental noise and repetitive resting state potentials that constitute redundant, task-irrelevant information [1][2]. The key features provide discriminative cues for emotion recognition, while the redundant information may obscure or interfere with the model's perception of sudden emotional changes. Therefore, the gating mechanism introduces learnable weight coefficients during feature extraction to dynamically adjust the response strength of each channel and time step, allocating higher weights at emotional change points to amplify discriminative signals and allocating lower weights in resting state or noisy segments to suppress interference. In this way, the model focuses on the most emotionally informative dynamic features, thereby improving recognition accuracy.
>
> 2.  **Bidirectional state-space modelling: Forward and backward features are constructed in parallel to capture richer contextual relationships.**
>
>     >I. **Bidirectional State-Space Modeling Along the Temporal Dimension:** Emotion recognition is inherently a sequential modeling task, since emotional states emerge from temporally dependent dynamics rather than isolated points[3]. Unidirectional modeling often struggles to capture strong temporal correlations, such as lagged or precursory signals, leading to overlooked emotional cues[4]. To address this, BiMamba first applies linear mapping, convolution, and activation for initial local feature extraction, then deploys forward and backward state-space model SSM branches in parallel along the time dimension. This bidirectional modeling perceives complete contextual dependencies from both temporal paths, enhancing the model's understanding and discrimination of cross-time emotional dynamics.
>
>     >II. **Bidirectional Cross-Modal Interaction Along the Channel Dimension:** Emotional states commonly exhibit correlations across multiple physiological modalities, which are reflected at the channel level. For instance, increases in EEG $\theta$- and $\delta$-band power often exhibit positive Pearson correlations with galvanic skin response (GSR) conductance peaks and negative correlations with electrocardiogram (ECG) heart-rate variability during emotional arousal[5]. In the forward branch, the representation of the $i$-th modality references the hidden states of preceding modalities from 1 through $i-1$. In the backward branch, it is supplemented by the hidden states of subsequent modalities from $M$ down to $i+1$, where $M$ is the total number of modalities. This bidirectional information flow enables each modality to leverage context from earlier modalities and complete its features using later ones, generating complementary expressions for the same modality and strengthening inter-modal associations and fusion.
>
> 3.  **Linear projection and residual connections: Integrate bidirectional features and alleviate the vanishing gradient problem.**
>
>     >We first fuse the forward feature $h_i^\rightarrow$ and backward feature $h_i^\leftarrow$ by averaging, then apply a linear mapping to merge features and perform dimensionality transformation. Finally, a residual connection is introduced to alleviate gradient vanishing.
>
> *Manuscript revision:* We have incorporated the above design thoughts into "3.2.1 Intra-modal BiMamba Module" and "3.2.2 Inter-modal BiMamba Module" to aid reader comprehension.
>
> ---
> ---
>
> # References
> [1] Barradas, I. et al. *Dynamic emotion intensity estimation from physiological signals*. PLOS ONE, 2025.
>
> [2] Brake, N., Duc, F., Rokos, A., Arseneau, F., Shahiri, S., Khadra, A., & Plourde, G., *A neurophysiological basis for aperiodic EEG and the background spectral trend*, Nature Communications, 2024.
>
> [3] Hofmann, S.M., Klotzsche, F., Mariola, A., Nikulin, V., Villringer, A., & Gaebler, M.. *Decoding subjective emotional arousal from EEG during an immersive virtual reality experience*. eLife, 2021.
>
> [4] Zhang X, Zhang Q, Liu H, et al. *Mamba in speech: Towards an alternative to self-attention*. IEEE Transactions on Audio, Speech and Language Processing, 2025.
>
> [5] Miranda-Correa JA, Abadi MK, Sebe N, Patras I. *AMIGOS: A dataset for affect, personality and mood research on individuals and groups*.  IEEE Transactions on Affective Computing, 2018.

---

> > ### Author Response · Authors · 2025-08-07
> > **Follow-up to Reviewer krBP**
> >
> > Dear Reviewer,
> >
> > ---
> >
> > We hope that the above clarifications have sufficiently addressed your concerns, and we would greatly appreciate your feedback.
> >
> > Thank you for your time and effort in reviewing our paper.
> >
> > ---
> >
> > Best Regards, Authors of Paper ID 8789

---

### Note · Authors · 2025-08-14

We have revised the manuscript based on the reviewers’ comments, with updates and summaries in three parts: **Manuscript update**, **Responses to reviewer concerns**, and **Our contributions**:


# Manuscript update
- **Clarified BiMamba design:** Summarized workflow and design rationale.
- **Enhanced TTA description:** Detailed two-level entropy-based filtering and its synergy with mutual information sharing.
- **Specified focus on partial missingness:** Explained rarity of complete modality missing in real-world data.
- **Explained modality contribution:** Linked to auxiliary loss weights and mutual-information hyperparameters.
- **Added hyperparameter analysis:** Described selection criteria (entropy threshold, smoothing factor).
- **Expanded baselines:** Included Mamba–Transformer hybrids with experiments.
- **Reorganized structure:** Added “Experimental Overview” and moved key ablations and hyperparameter studies to the main text.
- **Improved presentation:** Updated Figure 1 and added entropy-filtering illustration.

---

# Responses to reviewer concerns
 - **Reviewer krBP:** Clarified the design thoughts of BiMamba.
 - **Reviewer TeXe:** Clarified novelty concerns and added SSM–Transformer hybrid baselines.
 - **Reviewer RRS6:** Explained hyperparameter choices, including entropy threshold, smoothing factor, and iteration count; clarified why complete modality missingness is not considered; and highlighted BiMamba’s ability to capture inter-modal correlations.
 - **Reviewer Z1w5:** Clarified the two-level entropy-based sample filtering mechanism; improved Figure 1; and explained the number of iterations and threshold impact.

---

# Our contributions
- Designed a multimodal BiMamba network with intra-modal long-range dependency modeling and inter-modal correlation capture.
- Proposed a multimodal TTA to mitigate distribution shift from missing data.
- Achieved SOTA results on two multimodal emotion datasets, ensuring robust and accurate emotion recognition for clinical and everyday use.

We sincerely thank the reviewers and the AC for their valuable feedback, which helped us improve the technical details, experiments, and presentation, and further clarify the method and rationale to address the concerns.

---

### Decision · Program_Chairs · 2025-09-17

**Decision:**

Accept (poster)

**Comment:**

This work addresses two challenges in multimodal emotion recognition from physiological signals: (1) modeling long-range intra-modal dependencies and inter-modal correlations, and (2) handling distribution shifts and missing data at test time. The paper presents a new model (a multimodal bidirectional Mamba -BiMamba- network with test-time adaptation -TTA-) that achieves state-of-the-art performance on the DEAP and MAHNOB-HCI datasets.

All the reviewers are positive about the paper. The reviewers value both the innovative use of BiMamba and TTA to address the two challenges of emotion recognition from physiological signals studied in the paper, and the detailed experimental validation (including ablation studies, iterative TTA analysis, Grad-CAM, and t-SNE visualizations). Based on these considerations we recommend to accept the paper.